# Hangul Fonts Dataset: a Hierarchical and Compositional Dataset for Investigating Learned Representations

**Jesse A. Livezey**[1,2]        **Ahyeon Hwang**[3]        **Jacob Yeung**[1]

**Kristofer E. Bouchard**[1,2,4,5]

[1]Biological Sciences and Engineering Division, Lawrence Berkeley National Laboratory, Berkeley, CA
[2]Redwood Center for Theoretical Neuroscience, University of California, Berkeley, CA
[3]Mathematical, Computational and Systems Biology, University of California, Irvine, CA
[4]Helen Wills Neuroscience Institute, University of California, Berkeley, CA
[5]Computational Research Division, Lawrence Berkeley National Laboratory
Berkeley, CA
{jlivezey, kebouchard}@lbl.gov, ahyeon.hwang@uci.edu, jacobyeung@berkeley.edu

## Abstract

Hierarchy and compositionality are common latent properties in many natural and scientific datasets. Determining when a deep network's hidden activations represent hierarchy and compositionality is important both for understanding deep representation learning and for applying deep networks in domains where interpretability is crucial. However, current benchmark machine learning datasets either have little hierarchical or compositional structure, or the structure is not known. This gap impedes precise analysis of a network's representations and thus hinders development of new methods that can learn such properties. To address this gap, we developed a new benchmark dataset with known hierarchical and compositional structure. The Hangul Fonts Dataset (HFD) is comprised of 35 fonts from the Korean writing system (Hangul), each with 11,172 blocks (syllables) composed from the product of initial, medial, and final glyphs. All blocks can be grouped into a few geometric types which induces a hierarchy across blocks. In addition, each block is composed of individual glyphs with rotations, translations, scalings, and naturalistic style variation across fonts. We find that both shallow and deep unsupervised methods only show modest evidence of hierarchy and compositionality in their representations of the HFD compared to supervised deep networks. Supervised deep network representations contain structure related to the geometric hierarchy of the glyphs, but the compositional structure of the data is not evident. Thus, HFD enables the identification of shortcomings in existing methods, a critical first step toward developing new machine learning algorithms to extract hierarchical and compositional structure in the context of naturalistic variability.

## 1 Introduction

Advances in machine learning, and representation learning in particular, have long been accompanied by the creation and detailed curation of benchmark datasets [1–5]. Often, such datasets are created with particular structure believed to be representative of the types of structures encountered in the world. For example, many image datasets have varying degrees of hierarchy and compositionality,

Submitted to the 35th Conference on Neural Information Processing Systems (NeurIPS 2021) Track on Datasets and Benchmarks. Do not distribute.

as exemplified by parts-based decompositions, learning compositional programs, and multi-scale representations [6–8]. In contrast, synthetic image datasets often have known, (at least partial) factorial latent structure [9–11]. Having a detailed understanding of the structure of a dataset is critical to interpret the representations that are learned by any machine learning algorithm, whether linear (e.g., independent components analysis) or non-linear (e.g., deep networks). Learned representations can be used to understand the underlying structure of a dataset. Indeed, one of the desired uses of machine learning in scientific applications is to learn latent structure from complex datasets that provide insight into the data generation process [12–14]. Understanding how learned representations relate to the structure of the training data is an area of active research [15–18].

Benchmark image datasets such as MNIST (Fig 1A) and CIFAR10/100 [2, 19] enabled research into early convolutional architectures. Large image datasets like ImageNet (Fig 1B) and COCO [3, 20] have fueled the development of networks that can solve complex tasks like pixel-level segmentation and image captioning. Although these datasets occasionally have known semantic hierarchy (ImageNet classes are derived from the WordNet hierarchy [3, 21]) or labeled attributes which may be part of a compositional structure (attributes like "glasses" or "mustache" in the CelebA dataset [22]), the overall complexity of these images prevents a quantitative understanding of how the hierarchy or compositionality is reflected in the data or deep network representations of the data. On the other hand, synthetic benchmark datasets such as dsprites (Fig 1C), and many similar variations [9–11, 23], have known factorial latent structure [24]. However, these datasets typically do not have (known) hierarchy or compositionality. Thus, benchmark datasets, which have known hierarchical and compositional structure with naturalistic variability, are lacking.

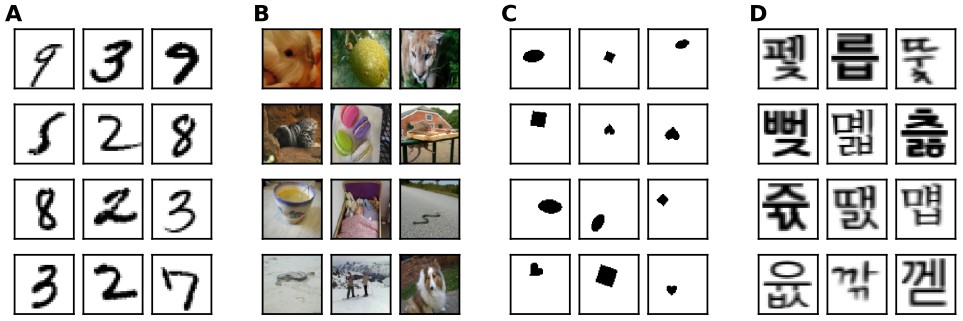

Figure 1: **Ground-truth hierarchy and compositionality are lacking in benchmark machine learning datasets. A** Samples from the MNIST dataset. **B** Samples from the ImageNet dataset. **C** Samples from the dsprites dataset. **D** Samples from the Hangul Fonts Dataset.

Machine learning and deep learning methods have been applied to a variety of handwritten and synthetic Hangul datasets with a focus on glyph recognition applications, font generation, and mobile applications [25–30]. HanDB is an early handwritten Hangul dataset [31] and contains approximately 100 samples of each of the 2350 most commonly used blocks. The similarly named Hangul Font Dataset packages a number of open fonts for potential machine learning applications with a focus on the vectorized contour information for the blocks rather than understanding the latent structure of the blocks [32]. As far as we are aware, the Hangul Fonts Dataset presented here is the only Hangul dataset that includes compositional and hierarchical annotations.

A number of methods have been proposed to uncover "disentangled" latent structure from images [6, 24, 33–44] and understand hierarchical structures in data and how they are learned in deep networks [15, 45]. For datasets where the form of the generative model is not known, deep representation learning methods often look for factorial or disentangled representations [33–35, 46, 47]. While factorial representations are useful for certain tasks like sampling [24], they do not generally capture hierarchical or compositional structures. Deep networks can learn feature hierarchies, wherein features from higher levels of the hierarchy are formed by the composition of lower level features. The hierarchical multiscale RNN captures the latent hierarchical structure by encoding the temporal dependencies with different timescales on for character-level language modelling and handwriting sequence generation tasks [48]. Deep networks have been shown to learn acoustic, articulatory, and visual hierarchies when trained on speech acoustics, neural data recorded during spoken speech syllables, and natural images, respectively [49–52]. Developing methods to probe representations for

hierarchical or compositional structures is important to develop in parallel to benchmark machine learning datasets.

In this work, we present the new Hangul Fonts Dataset (HFD) (Fig 1D) designed for investigating hierarchy and compositionality in representation learning methods. The HFD contains a large number of data samples (391,020 samples across 35 fonts), annotated hierarchical and compositional structure, and naturalistic variation. Together these properties address a gap in benchmark datasets for deep learning, and representation learning research more broadly. To give examples of the potential use of the HFD, we explore whether typical deep learning methods can be used to uncover the underlying generative model of the HFD. We find that deep unsupervised networks do not recover the hierarchical or compositional latent structure, and supervised deep networks are able to partially recover the hierarchy latent structure. Thus, the Hangul Fonts Dataset will be useful for future investigations of representation learning methods.

## 2 The Hangul Fonts Dataset

The Korean writing system (Hangul) was created in the year 1444 to promote literacy [53]. Since the Hangul writing system was partially motivated by simplicity and regularity, the rules for creating "blocks" are regular and well specified. The Hangul alphabet consists of "glyphs" broken into 19 initial glyphs, 21 medial glyphs, and 27+1 final glyphs (including no final glyph) which generate $19 \times 21 \times 28 = 11,172$ possible combinations of glyphs which are grouped into initial-medial-final (IMF) blocks. Not all blocks are used in the Korean language, however all possible blocks were generated for use in this dataset. The Hangul Fonts Dataset (HFD) uses this prescribed structure as annotations for the image of each block. The dataset consists of images of all blocks drawn in 35 different open-source fonts from [54–57] for a total of 391,020 annotated images. See Appendix C for detailed definitions of blocks, glyphs, and atoms and their linguistic meaning.

Each Hangul block can be annotated most simply as having initial, medial, and final (IMF) independent generative variables which can be represented as IMF class labels associated with each block. In addition, there are variables corresponding to a geometric hierarchy and variables corresponding to compositions of glyphs. The hierarchical variables are induced by the geometric layout of the blocks. There are common atomic glyphs used across the initial, medial, and final glyph positions (after a set of possible translations, rotations, and scalings) [58]. The compositional variables indicate which atomic glyphs are used for each block (in a "bag-of-atoms" representation). Together, these different descriptions of the data facilitate investigation into what aspects of this known structure representation learning methods will learn when trained on the HFD.

### 2.1 The structure of a block: hierarchy and compositionality

There are geometric rules for creating a block from glyphs. The initial glyph is located on the left or top of the block as either single or double glyphs (ㄱ or ㄲ in Fig 2A). There are 5 possible medial glyph geometries: below, right-single, right-double, below-right-single, or below-right-double (ㅗ, ㅏ, ㅔ, ㅘ, or ㅞ in Fig 2A). The final glyph is at the bottom of the block as single, double, or absent glyphs (ㄱ or ㄳ in Fig 2A). Grouping the blocks by the 30 geometric possibilities together induce a 2-level hierarchy based on their IMF class labels. The geometric variables describe the coarse layout (high level) of a block which is shared by many IMF combinations (low level) (Fig 2B and C, bottom and middle levels). Additionally, the 30 geometric categories can be split into their initial, medial or final geometries (Fig 2B and C, bottom and middle levels). The geometric context of a glyph can change the style of the glyph within a block for a specific font, which is relevant for the representation analysis in Section 3. The medial glyph geometry can have a large impact on how an initial glyph is translated and scaled in the block. Similarly, the final geometry can impact the scaling of the initial and medial glyphs. These contextual dependencies can be searched for in learned representations of the data. For example, a supervised deep network trained to predict the initial glyph class may use information from the medial geometry early in the network but then eventually discard that information when predicting the initial glyph class.

Since each block is composed of initial, medial, and final glyphs, the blocks can also be annotated with compositional features. There are a base set of atomic glyphs (atoms) from which all IMF glyphs are created (Fig 3A, Atom row). Then, one initial, one medial, and one final glyph are composed into a block (Fig 3A, IMF and Block rows). In this view, each block is built from a composition of a

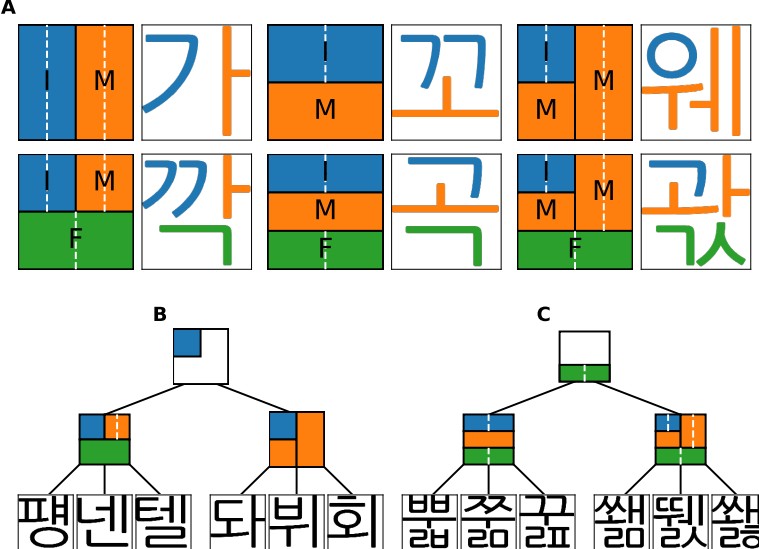

Figure 2: **Hierarchy in the Hangul Fonts Dataset. A, Hierarchy:** Each block can be grouped by the initial, medial, and/or final geometry. Block geometry and example blocks are shown. Blue indicates the possible locations of initial glyphs, orange indicates the possible locations of medial glyphs, and green indicates the possible locations of final glyphs. A white dashed line indicates that either a single or double glyph can appear. **B, C, Example hierarchies:** The bottom row of the hierarchy are individual blocks. Each triplet of blocks fall under one of the geometric categories from **A** (middle row) which defined the 2-level hierarchy. Then, a third level can be defined for initial, medial, or final geometric categories (top row).

base set of atomic glyphs potential composed with a rotation which are then laid out according to the geometric rules. The underlines in the Atom and IMF rows of Fig 3A correspond to inclusion in the final colored blocks in the bottom row. In this paper, for comparisons with learned representations, the composition features are encoded in 2 ways (although the full structure is available in the dataset). The first is a "bag-of-atoms mod rotations" feature where each block is given a vector of binary features which contains a 1 if the block contains at least one atom from the top row of Fig 3A in any position with any rotation and a 0 otherwise (16 total features). The second is a similar "bag-of-atoms" feature where the same atomic glyph with different rotations are given different feature elements (24 features). These two feature sets do not encode the complete compositional structure, but they are amenable to common representation comparison methods.

These three sets of variables—IMF class labels, hierarchy class labels, and bag-of-atoms binary features—are not independent of each other. For example, training on the Initial class label may automatically structure the learned representations around the Initial Geometry labels since they are partially correlated. However, it is not clear whether this provides an upper (or lower) bound for the expected structure of related variables in the representation. For example, if a network is trained on the Initial classes and learns a highly clustered representation for each class, it is not guaranteed the network will always put classes that share Initial Geometry hierarchy close to each other in the learned representations. Indeed, this is a hypothesis we are hoping to test with this dataset across representation learning methods. This could result in clustering accuracies lower than what was expected based on the label correlations. Similarly, the network could perfectly group Initial class representations around their Initial Geometry labels and the clustering accuracy would be set by the Initial accuracy with some conversion to account for different numbers of classes.

The size and shape of a glyph can change within a font depending on the context. Some of these changes are consistent across fonts and stem from the changing geometry of a block with different initial, medial, or final contexts (Fig 2). Different types of variations such as rotation, translation, and more naturalistic style variations arise in the dataset (Fig 3B). Glyphs can incorporate different rotations, scalings, and translation during composition into a block (Fig 3B, left 3 sets). There are variations across fonts due to the nature of the design or style of the glyphs. These include the style of

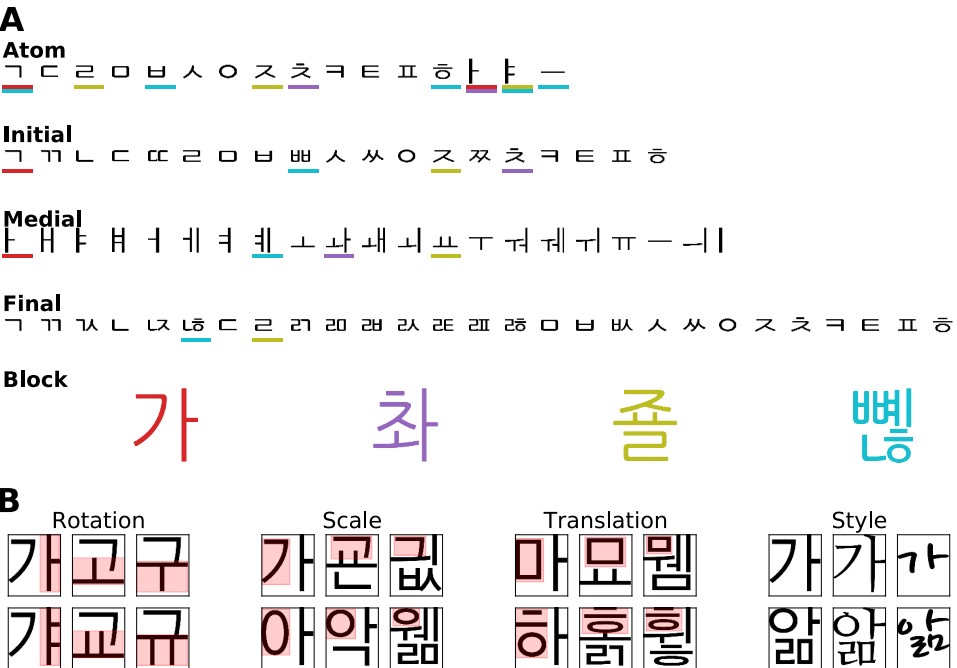

Figure 3: **Composition and variation in the Hangul Fonts Dataset. A, Composition:** Each block is composed of a set of atomic glyphs. The Atom row shows the atomic set of glyphs when scale, translations, and rotations are modded out. The Initial, Medial, and Final (IMF) rows show all IMF glyphs. The Block row shows four example blocks with different types of structure. The color of the block is used to underline the IMF glyphs that compose the block and Atoms that compose the IMFs. **B, Variability:** Two example glyphs (rows) across three different IMF contexts (columns) are shown for each type of variation. **Rotation:** Left-most block is rotated once counterclockwise in the next block, then twice counterclockwise in the final block. **Scale:** Size of initial glyph decreases from left to right as highlighted in red. **Translation:** Highlighted glyph takes on various shapes as it is translated to different regions of the block. **Style:** Less to more stylized from left to right.

glyphs which can vary from clean, computer font-like fonts to highly stylized fonts which are meant to resemble hand-written glyphs (Fig 3B, rightmost set). Line thickness and the degree to which individual glyphs overlap or connect also vary. This variation is specific to a font and is based on the decision the font designer made, analogous to hand-written digits (i.e., MNIST). These types of variation are the main source of naturalistic variation in the dataset since they cannot be exactly described, but could potentially be modeled [7, 44].

## 2.2  Generating the dataset

We created a text file for the 11,172 blocks using the Unicode values from [59]. We then converted the text files to an image file using the `convert` utility [60] and font files. The image sizes were different across blocks within a font, so the images were resized to the max image size across blocks in the font. As the image sizes of blocks were also different across fonts, the blocks were resized to the median size across fonts. Individual images for the initial, medial, and final glyphs are included, when available. The exact scripts used to generate the dataset, a Dockerfile which can be used to recreate or extend the HFD, curated open fonts, and pseudo-code for the generation process are provided (see Appendix A). Further summary statistics for the dataset can be found in Appendix B.

## 3  Searching for hierarchy and compositionality in learned representations

Both shallow and deep learning models create representations (or transformations) of the input data. Methods like Principal Components Analysis (PCA) produce linear representations and Nonnegative Matrix Factorization (NMF) produces a shallow nonlinear representation through inference in a linear

generative model, and deep networks produce an increasingly nonlinear set of representations for each layer. Here, we compare the learned representation in unsupervised shallow methods, deep variational autoencoders, and deep feedforward classifiers. We consider whether the learned representations are organized around any of the categorical labels and hierarchy variables with an unsupervised KMeans analysis. Then, we investigate whether the hierarchy or compositionality variables can be decoded with high accuracy from few features in the representations.

It is desirable that deep network representations can be used to recover the generative variables of a dataset. However, it is currently not known whether deep network representations are typically organized around generative variables. In order to understand this, we test whether the latent hierarchical structure of the Hangul blocks is a major component of the learned representations using unsupervised clustering of the representations. We compare the hierarchy geometry classes from Fig 2A to KMeans clusterings of the test set representations (where $k$ is set to the number of class in consideration, for more details, see Appendix D). For the shallow and deep unsupervised methods (Fig 4A and B), we find that the medial label and geometry, final label, and all_geometry variables are all marginally present ($0 <$ normalized accuracy $\leq 0.25$, see Section 4 for definition) in the representations. The other variables are not recovered by the unsupervised methods (normalized accuracy $\approx 0$). This shows that while VAE variants may be able to disentangle factorial structure in data, they are not well suited to extracting geometric hierarchy from the HFD with high fidelity.

In contrast (and unsurprisingly), supervised deep networks cleanly extract and recover the label they are trained on (Fig 4C-E, first 3 columns) with increasing accuracy across layers (Norm. acc. $> 0.25$). When trained on the initial label, the initial, medial, and all_geometry variables can all be marginally recovered, highlighting the contextual dependence of the initial glyph on the medial geometry. The medial_geometry variable can be decoded with accuracy significantly above chance across all layers ($p < .01$, 1-sample t-test). However, the normalized accuracy drops from about 0.22 in the first layer to less than .01 by the last layer. This indicates that although the network may be using the medial geometry context in the early layers, it is compressed out of the representation by the final layers. The initial geometry is not present in the first 2 layers, but becomes marginally present in the final layers. When trained on the medial labels, the medial geometry is present with high accuracy and the all geometries labels are marginally present. When trained on the final labels, the final geometry becomes present by the last 2 layers. There is a small amount of interaction with the medial geometry, but it is not as large as the initial-medial interaction. There are several mean normalized accuracies that are less than zero. Although it is potentially interesting that it only occurs for Initial Geometry, the negative values all have pvalues $> .01$ (1 sample t-test) and some are not significantly different from 0. In addition, the significant differences from 0 are relatively small. Furthermore, inspecting the per-fold accuracies shows that it was just one or two of the 7 folds that had a larger below chance accuracy. Given this, we would attribute this to statistical fluctuations or overfitting rather than a meaningful signal. These results indicate that supervised deep networks do learn representations that mirror aspects of the hierarchical structure of the dataset that are most relevant for the task, and generally do not extract non-relevant hierarchy information.

Understanding whether deep network representations tend to be more distributed or local is an open area of research [17, 61, 62]. We investigated whether deep networks learn a local representation by training sparse logistic regression models to predict the latent hierarchy and compositionality variables from the representations (Fig 5). If the representation of a hierarchy or compositionality variable is present and simple (linear), we would expect the normalized accuracy to be high (near 1 on the y-axis of the plots in Fig 5). If a representation of a variable is "local", we would expect the variable to be decoded using approximately the same number of features as it has dimensions (near $10^1$ on the x-axis of Fig 5) and "distributed" representation to have a much higher ratio. To test this, we compare these two measures across models and target variables and also across layers for the supervised deep networks.

We find that unsupervised ($\beta$-)VAEs (Fig 5A) learn consistently distributed representations of the latent variables (typically 30-60x more features than the variable dimension are selected). In terms of the prediction accuracy, the cross validated $\beta$-VAEs tend to have higher accuracy across variables than the VAE and the $\beta$-VAE selected for traversals, although there is a fair amount of heterogeneity. For supervised deep networks (Fig 5B-D), the supervision variable (initial, medial, final, respectively), has high accuracy across layers, and moves from a more distributed to a more local representation at deep layers. For the initial and medial labels, the medial geometry can also be read out with high accuracy and an increase in localization across layers. The initial geometry is not read out with

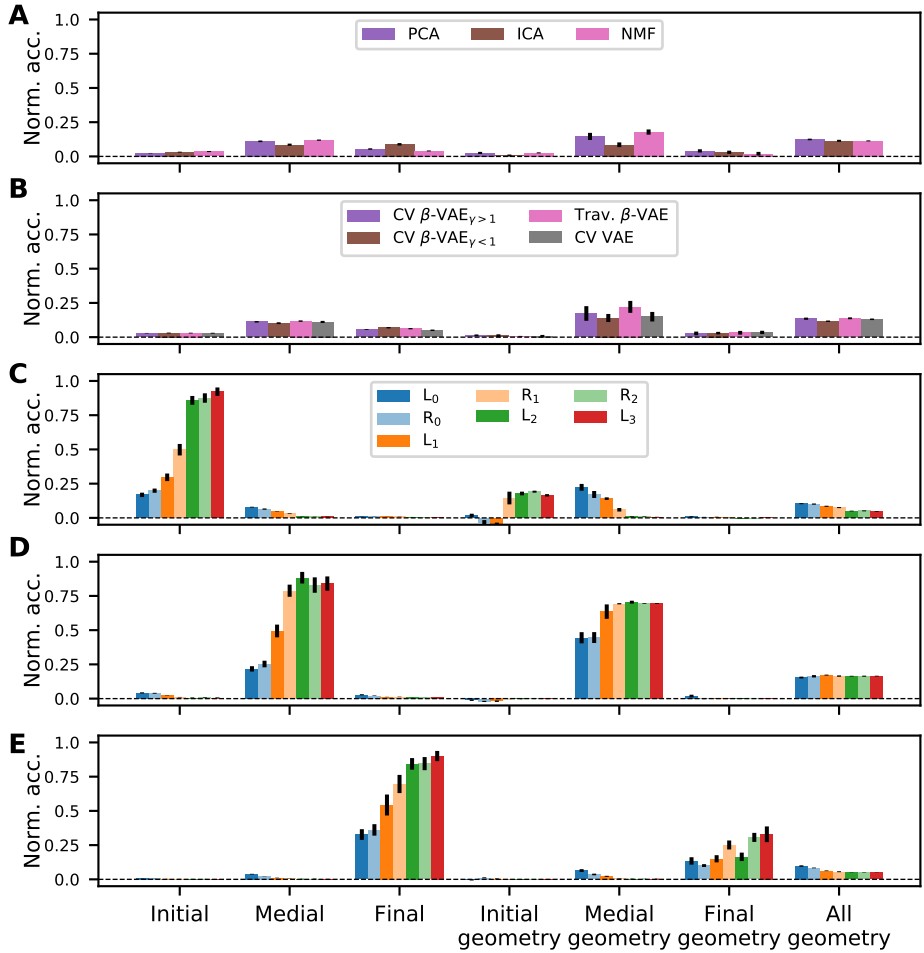

Figure 4: **Representation learning methods partially recover the geometric hierarchy.** Normalized clustering accuracy $\pm$ s.e.m. is shown across training targets, latent generative variables, layers (L is the linear part, R is after the ReLU), and model types. **A** Normalized clustering accuracies for representations learned with unsupervised linear models. **B** Normalized clustering accuracies for representations learned with various deep VAE models. **C-E** Normalized clustering accuracies for deep representations trained to predict the initial, medial, and final label, respectively.

high accuracy in the initial and medial label networks, and the final geometry variable can only be predicted well for the final label network. The all_geometry variable can be predicted at marginal accuracy for all networks. The compositional Bag-of-Atoms (BoA) features cannot be predicted well (often at or below chance) for any network and the BoA mod rotations can only be read out with marginal accuracy for the initial label network. These results suggest that standard, fully-connected deep networks do not typically learn local representations for variables except for those they are trained on (and correlated variables).

## 4 Methods

### 4.1 Representation learning methods

Principal Component Analysis (PCA), Independent Component Analysis (ICA), and Non-negative Matrix Factorization (NMF) from Scikit-Learn [63] were used to learn representations from the data. These methods were all trained with 100 components which is at least 3-times larger than any of

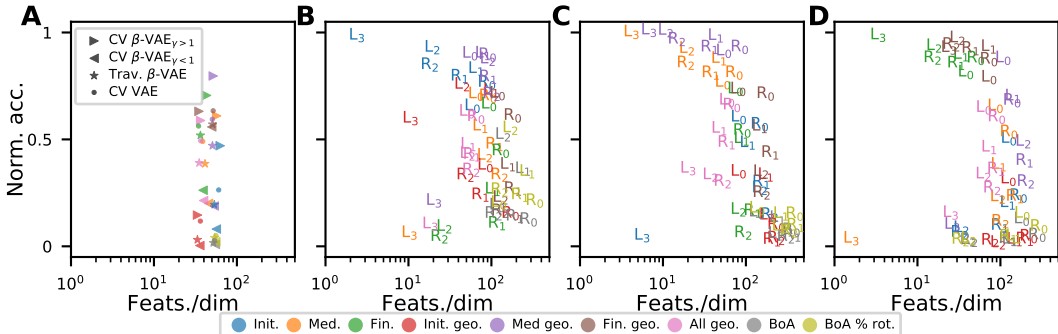

Figure 5: **Hierarchy and compositionality are not typically represented locally in deep networks.** Held-out logistic regression normalized accuracy is shown versus the ratio of the number of features selected to the variable dimensionality. Color indicates latent variable type. **A**: Results from the VAE model variants. Shape is model type. **B-D:** Results from supervised deep networks trained on the initial, medial, and final tasks, respectively. Letters in correspond to the layers from Fig 2.

Figure 6: **Disentangled reconstructions from $\beta$-VAE.** Latent traversals of a single latent variable. The left column is the input image, middle columns are the traversals, and right column is the block the traversals appear to morph into. **A, Initial Across Fonts:** First four rows are similar traversals of an initial glyph from one block across increasingly naturalistic fonts. Final row is an entangled traversal between initial and final glyphs. **B, Final Across Fonts:** First four rows are similar traversals of a final glyph from one block across different fonts. Final row is an entangled traversal between initial, medial, and final glyphs. **C, Final Across Blocks:** First two rows are similar traversals of a final glyph from blocks (with the same hierarchy) in the same font. Third row is a traversal of a final glyph from a block (with a different hierarchy). Fourth row is an entangled traversal between initial and final glyphs. Final row shows an entangled traversal of medial and final glyphs.

the latent generative variables under consideration. The models were trained on the training and validation sets and the representation analysis was on the test set.

Variational autoencoders (VAEs) learn a latent probabilistic model of the data they are trained on. The $\beta$-VAE is a variant of a VAE which aims to learn disentangled latent factors [34, 35] by trading off the reconstruction and KL-divergence terms with a factor different than 1. We implement the $\beta$-VAE from Burgess et al. [35], which encourages the latent codes to have a specific capacity. We experiment with both $\beta > 1$ from [35] as well as $\beta < 1$ from [64, 65]. $\beta$-VAE networks with convolutional and dense layers were trained on the dataset. 100 sets of hyperparameters were used for training the $\beta$-VAEs. The hyperparameters and their ranges are listed in Appendix E. In order to cross-validate the networks, we checked if the same blocks across fonts are nearest neighbors in the latent space. For each block in each font, the nearest neighbor is found. If the neighbor has the same label as the block, we assign an accuracy of 1, otherwise 0. This is averaged across all blocks and pairs of fonts in the validation set. The model with the best cross-validation accuracy for each label was chosen and the downstream analysis was done on the test set latent encodings. We also cherry-picked networks which had interpretable latent traversals (Fig 6).

Fully-connected networks with 3 hidden layers were trained on one of the initial, medial, or final glyph variables. For each task, 100 sets of hyperparameters were used for training. The hyperparameters and their ranges are listed in Appendix E. The model with the best validation accuracy was chosen and the downstream analysis was done on the test set representations (test accuracies reported in

257 Appendix B). Code for training the networks and reproducing the figures will be posted publicly.
258 Deep networks representation analysis was partially completed on the NERSC supercomputer. All
259 deep learning models were trained using PyTorch [66] on Nvidia GTX 1080s or Titan Xs.

260 To compare accuracies (and chance accuracies) across models with differing numbers of classes
261 (between 2 and 30), we 0-1 normalize the accuracies across models to make comparisons more clear.
262 Specifically, for a model with accuracy $= a$ and chance $= c$, we report Norm. acc. $= \frac{a-c}{1-c}$ which is 0
263 when $a = c$ and is 1 when $a = 1$, independent of the number or distribution of classes.

## 4.2 Generative structure recovery from representation of the data

265 The 35 fonts were used in a 7-fold cross validation loop for the machine learning methods. The fonts
266 were randomly permuted and then 5 fonts were used for each of the non-overlapping validation and
267 test sets. The analysis of representations was done on the test set representations. For the supervised
268 deep networks, the Kmeans clustering analysis and sparse logistic regression analysis were applied to
269 the activations of every layer both before and after the ReLU nonlinearities. For the unsupervised
270 VAEs, they were applied to samples from the latent layer. The logistic regression analysis was not
271 applied to the linear representations.

272 Clustering a representation produces a reduced representation for every datapoint in an unsupervised
273 way. If one chooses the number of clusters to be equal to the dimensionality or number of classes the
274 generative variables has, then they can be directly compared (up to a permutation). We cluster the
275 representations with KMeans and then find the optimal alignment of the real and clustered labels (see
276 Appendix D for more details). We then report the normalized accuracy of this labeling across training
277 variables, layers, and hierarchy variables.

278 Sparse logistic regression attempts to localize the information about a predicted label into a potentially
279 small set of features. To do this, we used logistic regression models fit using the Union of Intersection
280 (UoI) method [67, 68]. The UoI method has been shown to be able to fit highly sparse models
281 without a loss in predictive performance [69]. We report the normalized accuracy and mean number
282 of features selected divided by the number of features or classes across training variables, layers, and
283 hierarchy variables. For this analysis, 2 new training and testing sub-splits were created from the
284 representations on the original test set that was held out during deep network training.

## 5 Discussion

286 The Hangul Fonts Dataset (HFD) presented here has hierarchical and compositional latent structure
287 that allows each image (block) to have ground-truth annotations, making the HFD well suited for
288 deep representation research. Using a set of unsupervised and supervised methods, we are able to
289 extract a subset of the variables from the representations of deep networks. Several VAE variants
290 have relatively poor variable recovery from their latent layers, while supervised deep networks have
291 clear representation of the variables they are trained on and interacting variables. Understanding how
292 to better recover such structure from deep network representations will broaden the application of
293 deep learning in science.

294 In many scientific domains like cosmology, neuroscience, and climate science, deep learning is being
295 used to make high accuracy predictions given growing dataset sizes [50, 70–72]. However, deep
296 learning is not commonly used to directly test hypotheses about dataset structure. This is partially
297 because the nonlinear, compositional structure of deep networks, which is conducive to high accuracy
298 prediction from complex data, is not ideal for interrogating hypotheses about data. In particular, it
299 is not generally known how the structure of a dataset influences the learned data representations or
300 whether the structure of the dataset can be "read-out" of the learned representations. Understanding
301 which dataset structures can be extracted from learned deep representations is important for the
302 expanded use of deep learning in scientific applications.

303 The HFD is based on a set of fonts which provide some naturalistic variation. However, the amount
304 of variation is likely much smaller than what would be found in a handwritten dataset of Hangul
305 blocks. One benefit to using fonts is that the dataset can be easily extended as new fonts are created.
306 To this end, we release the entire dataset creation pipeline to aid in future expansion of the HFD or the
307 creation of similar font-based datasets. A related limitation is that by including all possible blocks in
308 the datasets, a large fraction of the blocks in the HFD would almost never be found in natural writing

datasets. As is, the HFD could potentially bias machine learning applications which are applied to natural writing. To address this, the HFD could be subsampled to the relevant subset of blocks that are commonly used.

Another potential limitation and area of future work is determining how to encode variables like hierarchy and compositionality. In this dataset, there is a natural class-based encoding for the shallow geometry hierarchy. The Bag-of-Atoms composition encoding ignores structure that is potentially relevant for recovering compositionality (much like Bag-of-Words features discard potentially useful structure in natural language processing). The specific compositional and hierarchical structure in the HFD and the particular encodings used may not be applicable across all different types of compositionality or hierarchy, for instance some hierarchy may be fuzzy, rather than discrete and tree-like. Similarly, the analysis presented here is tailored to the particular structures present in the data. For example, the KMeans clustering analysis was applied to all variables with mutually-exclusive class structure, but could not be applied to the bag-of-atoms feature vectors. However, we hope that the HFD inspires more research into tools for extracting these features from learned representations.

In this work, relatively small fully-connected and convolutional networks were considered. However, these techniques can be applied to larger feedforward networks, recurrent networks, or networks with residual layers to understand the impact on learned representations. Understanding how proposed methods for learning factorial or disentangled representations [24, 33, 34, 40] impact the structure of learned representations is important for using deep network representations for hypothesis testing in scientific domains. Compared to disentangling [46], relatively little work addresses how to define and evaluate hierarchy and compositionality in learned representations. Furthermore, unsupervised or semi-supervised cross-validation metrics that can be used for model selection across a range of structure recovery tasks (e.g., disentangling, hierarchy recovery, compositionality recovery) are lacking.

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
