# OpenReview forum: "Hangul Fonts Dataset: a Hierarchical and Compositional Dataset for Investigating Learned Representations"
_NeurIPS.cc/2021/Track/Datasets_and_Benchmarks/Round1 — Submitted to NeurIPS 2021 Datasets and Benchmarks Track (Round 1)_

### Official Review · Reviewer_u1Si · 2021-07-01
**The paper introduces the Hangul fonts dataset and provides an interesting empirical investigation with respect to the hierarchical and compositional structure of the dataset. The contribution is solid, but discussion of related work could further strengthen the contribution. Concerns over public availability need to be clarified.**

**Rating:** 7
**Confidence:** 4

**Strengths:**

* The paper is generally well written and easy to follow.
* Hangul serves as a very intuitive writing system to allow a meaningful analysis of compositionality and hierarchy in ML models. Even for the reader who is unaware of the Korean language, it is easy enough to follow the underlying rules of generation.
* In my own humble opinion, analysing a dataset such as the presented one, could provide more meaningful insights than the typically conducted “disentanglement” analysis on dataset like Celeb, where models seem somewhat cherry picked and are labelled as disentangling when some concepts like “glasses” are empirically observed to be separated, while other factors remain densely entangled.
* Whereas I initially was lacking some crucial information (e.g. descriptions required for reproducibility), the majority of my open questions were clarified when going through the detailed appendix. The Appendix is a nice complement to the main body and provides a lot of additional information. This includes the dataset sheet.
* There is an interesting set of initial experiments that include both a DNN perspective, but also more traditional approaches like PCA, ICA or NMF. Now one the one hand, one could come to the conclusion that many of the presented insights are well known, such as the fact that training beta-VAEs does not actually result in desired disentanglement. However, I would personally argue that this is mainly a dataset paper, and actually seeing this analysis being done is positive, even if the outcome is somewhat expected, rather than just handwaving or referencing other papers. I believe most readers will appreciate the initial and rather thorough empirical analysis (including the additional content of the appendix)

**Weaknesses:**

In my opinion the paper has two weaknesses and one open major concern.

1. The related work section is focused on datasets like MNIST and ImageNet, but there is no mention or discussion of actual datasets surrounding the Korean language/Hangul. To the best of my knowledge, I personally cannot think of any dataset that has a similar compositionality in terms of generation at an atomic level, but there do exist multiple very interesting Hangul datasets already. E.g. one of them would be the HangulDB (and its various publications - see https://github.com/callee2006/HangulDB) with handwritten Hangul. Or IBM/Tensorflow’s hangul recognition (https://github.com/IBM/tensorflow-hangul-recognition).  The least the authors should do is cite prior works on Hangul databases and provide a discussion of related works on the topic. See also next point.
2. Unfortunately I am no linguist, but from what I understand, a good portion of the 11k combinations of syllables does not actually find practical use in words in the practical language. From a machine learning standpoint, the investigation of compositonality/hierarchy of the dataset of course remains interesting. However, it would be nice to have a better discussion of limitations/implications for the actual language and ML systems trained for practical tasks like recognition/translation. I believe this point would be okay, if related works (see point) were discussed in sufficient depth and differentiated first.
3. I find it concerning that the release of the dataset /code and licensing is written in future tense, with many explicit “TODO” placeholders in the appendix.  The authors write that the release is planned and licensing is currently pending institutional review. I honestly believe that this should have been done before submission. To the best of my understanding, the rationale behind the single-blind review process of the NeurIPS benchmarks and datasets track was to allow authors to make the datasets/resources publicly available already and sort these things out prior to the review process. Obviously, the strength of a dataset/benchmark paper lies in its public accessibility. I imagine the authors are naturally intending to do a public release (like the suggested Zenodo record). There already exists a GitHub link with code, but it would have been great to clarify potential licensing issues of code and dataset beforehand to make sure it is not too prohibitive (what if the paper gets accepted and the institutional review does not concur with the suggested license? Or the code never gets released). I will currently give the benefit of doubt, but this point should be clarified before a final decision is reached.
* As an addition to point 3: it remains unclear whether the code for the conducted investigation will be released as well? This should be clarified, as open-sourcing the code for the baseline analysis would strengthen the contribution. Lines 228 states “code for training the networks and reproducing the figures will be posted publicly”. Again, given that the review is single blind, this should have been done before, instead of promising to do this upon acceptance.

I am willing to raise my score if the related work section with respect to Hangul is improved and the public availability is addressed.


**Additional Feedback:**

As a minor remark, there’s several typos and sentences to be corrected.

Main paper:
* lines 79-21 “We find that deep unsupervised networks to do recover the hierarchical …”  -> ?
* line 142: availble -> available
* line 211: probabalistic ->  probabilistic
* line 226: are listen -> are listed
* lines 238-241: “For the supervised deep networks, the following methods were applied to the activations of every layer both before and after the ReLU nonlinearities. For the unsupervised ….” -> what methods?

Appendix:
* line 125: availble -> available
* lines 106 and 125 have a “TODO”. This should be replaced with the actual content.

**Clarity:**

The paper is well written and easy to follow. I did not find any paragraph that was unclear. There is a sufficiently detailed introduction to Hangul, so that I believe even the reader unaware of the language will be able to follow the idea.

**Correctness:**

I believe the dataset has generally been constructed in a sound way. The experiments seem rigorous, with statistical deviations reported and a meaningful set of chosen hyper-parameters, as reported in the appendix.

**Documentation:**

The dataset is reasonably well documented and a dataset sheet is provided. As a minor remark for improvement, the employed generation process could be included (e.g. in form of pseudo-code) in the appendix.

**Ethics:**

There are no ethical concerns.

**Relation To Prior Work:**

The paper has a rather detailed list of references with respect to models, the concept of disentanglement and the empirical analysis part. However, as mentioned under weaknesses, the reported related work with respect to Hangul datasets and their investigation in ML is lacking.

**Summary And Contributions:**

The paper introduces the Hangul fonts dataset, based on the fact that Korean Hangul enjoy a compositional structure. The dataset itself is of synthetic nature, based on a generative process that creates a total of up to 11k syllables under various fonts and several variations, such as rotation or style. To highlight the potential for investigation of how learned representations in (deep) machine learning are (in-)capable of capturing hierarchy/compositionality, an empirical analysis is showcased. The latter demonstrates the persisting gap in present unsupervised and supervised methods with respect to typically desired disentanglement of representations.

The initial review is leaning towards acceptance, *conditioned on the fact that the public availability of code/datasets and their licenses is clarified first, and the related work section is improved*.

----------------
**Update after authors' response**

I believe the authors have made updates that clarify some of the raised concerns (description of related work, licensing, accessibility, reproducibility with code).

Whereas some of the feedback is naturally still to be tackled in future work, I think the advantages of introducing this dataset to the community outweigh the potential remaining open points.

I have thus raised my score from 6 to 7.

---

> ### Author Response · Authors · 2021-07-14
> **Detailed review response**
>
> > The initial review is leaning towards acceptance, conditioned on the fact that the public availability of code/datasets and their licenses is clarified first, and the related work section is improved.
>
> We appreciate the constructive review and are happy to report that the licensing process is complete and the dataset generation and additional analysis and plotting code is being released under a Lawrence Berkeley National Labs BSD variant license (see https://github.com/BouchardLab/HangulFontsDatasetGenerator/blob/main/LICENSE.txt). Additionally, we have packaged the version of the code used in the manuscript into a Zenodo repository (https://zenodo.org/record/5104937)
>
> >Strengths:
>
> >- The paper is generally well written and easy to follow.
> >- Hangul serves as a very intuitive writing system to allow a meaningful analysis of compositionality and hierarchy in ML models. Even for the reader who is unaware of the Korean language, it is easy enough to follow the underlying rules of generation.
> >- In my own humble opinion, analysing a dataset such as the presented one, could provide more meaningful insights than the typically conducted “disentanglement” analysis on dataset like Celeb, where models seem somewhat cherry picked and are labelled as disentangling when some concepts like “glasses” are empirically observed to be separated, while other factors remain densely entangled.
> >- Whereas I initially was lacking some crucial information (e.g. descriptions required for reproducibility), the majority of my open questions were clarified when going through the detailed appendix. The Appendix is a nice complement to the main body and provides a lot of additional information. This includes the dataset sheet.
> >- There is an interesting set of initial experiments that include both a DNN perspective, but also more traditional approaches like PCA, ICA or NMF. Now one the one hand, one could come to the conclusion that many of the presented insights are well known, such as the fact that training beta-VAEs does not actually result in desired disentanglement. However, I would personally argue that this is mainly a dataset paper, and actually seeing this analysis being done is positive, even if the outcome is somewhat expected, rather than just handwaving or referencing other papers. I believe most readers will appreciate the initial and rather thorough empirical analysis (including the additional content of the appendix)
>
> We are grateful that the motivation for this dataset and example analysis was clear.
>
> >Weaknesses:
>
> >In my opinion the paper has two weaknesses and one open major concern.
>
> >1. The related work section is focused on datasets like MNIST and ImageNet, but there is no mention or discussion of actual datasets surrounding the Korean language/Hangul. To the best of my knowledge, I personally cannot think of any dataset that has a similar compositionality in terms of generation at an atomic level, but there do exist multiple very interesting Hangul datasets already. E.g. one of them would be the HangulDB (and its various publications - see https://github.com/callee2006/HangulDB) with handwritten Hangul. Or IBM/Tensorflow’s hangul recognition (https://github.com/IBM/tensorflow-hangul-recognition). The least the authors should do is cite prior works on Hangul databases and provide a discussion of related works on the topic. See also next point.
>
> This was an oversight on our part and we appreciate the pointers. We have added an additional paragraph in the introduction which situates the Hangul Fonts Dataset with respect to other datasets and studies which use the Hangul writing system. During this literature search, we also found and cite the similarly named Hangul Font Dataset (ref 32) which is based on a similar set of open source fonts. They focus on the vectorized contours that make up the blocks rather than the compositional and hierarchical structure of the blocks. It was released between the preprint of our manuscript and our submission to this dataset track.
> The paragraph reads (and see refs 25 - 32):
> “Machine learning and deep learning methods have been applied to a variety of handwritten and synthetic Hangul datasets with a focus on character recognition applications, font generation, and mobile applications [25-30]. HanDB is an early handwritten Hangul dataset [31] and contains approximately 100 samples of each of the 2350 most commonly used blocks. The similarly named Hangul Font Dataset packages a number of open fonts for potential machine learning applications with a focus on the vectorized contour information for the blocks rather than understanding the latent structure of the blocks [32]. As far as we are aware, the Hangul Fonts Dataset presented here is the only Hangul dataset that includes compositional and hierarchical annotations.”
>
> continued...

---

> > ### Author Response · Authors · 2021-07-14
> > **Detailed review response continued**
> >
> > >2. Unfortunately I am no linguist, but from what I understand, a good portion of the 11k combinations of syllables does not actually find practical use in words in the practical language. From a machine learning standpoint, the investigation of compositonality/hierarchy of the dataset of course remains interesting. However, it would be nice to have a better discussion of limitations/implications for the actual language and ML systems trained for practical tasks like recognition/translation. I believe this point would be okay, if related works (see point) were discussed in sufficient depth and differentiated first.
> >
> > Yes, it is correct that a majority of the 11k syllables are not used in the spoken/written language and you are correct to point out that this has implications for its direct use outside of compositionality/hierarchy research. On the other hand, if the desired distribution of blocks was available, this dataset could be appropriately subsampled.
> > We have expanded on this limitation in the Discussion section:
> > “A related limitation is that by including all possible blocks in the datasets, a large fraction of the blocks in the HFD would almost never be found in natural writing datasets. As is, the HFD could potentially bias machine learning applications which are applied to natural writing. To address this, the HFD could be subsampled to the relevant subset of blocks that are commonly used.”
> >
> > >3. I find it concerning that the release of the dataset /code and licensing is written in future tense, with many explicit “TODO” placeholders in the appendix. The authors write that the release is planned and licensing is currently pending institutional review. I honestly believe that this should have been done before submission. To the best of my understanding, the rationale behind the single-blind review process of the NeurIPS benchmarks and datasets track was to allow authors to make the datasets/resources publicly available already and sort these things out prior to the review process. Obviously, the strength of a dataset/benchmark paper lies in its public accessibility. I imagine the authors are naturally intending to do a public release (like the suggested Zenodo record). There already exists a GitHub link with code, but it would have been great to clarify potential licensing issues of code and dataset beforehand to make sure it is not too prohibitive (what if the paper gets accepted and the institutional review does not concur with the suggested license? Or the code never gets released). I will currently give the benefit of doubt, but this point should be clarified before a final decision is reached.
> >
> > These are all fair criticisms. We are happy to report that the code has been licensed under a Lawrence Berkeley National Labs BSD variant license (see https://github.com/BouchardLab/HangulFontsDatasetGenerator/blob/main/LICENSE.txt). Additionally, we have packaged the version of the code used in the manuscript into a Zenodo repository.
> >
> > >As an addition to point 3: it remains unclear whether the code for the conducted investigation will be released as well? This should be clarified, as open-sourcing the code for the baseline analysis would strengthen the contribution. Lines 228 states “code for training the networks and reproducing the figures will be posted publicly”. Again, given that the review is single blind, this should have been done before, instead of promising to do this upon acceptance.
> >
> > Again, this is a fair criticism. We have included the analysis scripts, modules, and plotting notebooks in the dataset repository under the `hangul_analysis` folder (https://github.com/BouchardLab/HangulFontsDatasetGenerator/tree/main/hangul_analysis).
> >
> > > Relation To Prior Work:
> >
> > >The paper has a rather detailed list of references with respect to models, the concept of disentanglement and the empirical analysis part. However, as mentioned under weaknesses, the reported related work with respect to Hangul datasets and their investigation in ML is lacking.
> >
> > We thank the reviewer for pointing this out and have expanded on Hangul related work in the introduction.
> > Documentation:
> >
> > >The dataset is reasonably well documented and a dataset sheet is provided. As a minor remark for improvement, the employed generation process could be included (e.g. in form of pseudo-code) in the appendix.
> >
> > We appreciate this suggestion and have included pseudo-code for the generation process in Appendix A.
> >
> > continued...

---

> > > ### Author Response · Authors · 2021-07-14
> > > **Detailed review response continued**
> > >
> > > >As a minor remark, there’s several typos and sentences to be corrected.
> > >
> > > >Main paper:
> > >
> > > >- lines 79-21 “We find that deep unsupervised networks to do recover the hierarchical …” -> ?
> > > >- line 142: availble -> available
> > > >- line 211: probabalistic -> probabilistic
> > > >- line 226: are listen -> are listed
> > > >- lines 238-241: “For the supervised deep networks, the following methods were applied to the activations of every layer both before and after the ReLU nonlinearities. For the unsupervised ….” -> what methods?
> > >
> > > >Appendix:
> > >
> > > >- line 125: availble -> available
> > > >- lines 106 and 125 have a “TODO”. This should be replaced with the actual content.
> > >
> > > We have corrected these typos/omissions in the main text and appendix.

---

> > > > ### Comment · Reviewer_u1Si · 2021-07-15
> > > > **Many points have been addresses, score has been raised. One remaining point**
> > > >
> > > > Thank you for your response and updates to the paper.
> > > >
> > > > Some of my key concerns have been addressed. As the authors themselves state, there is still some open points that would be good to include in the future. However, I believe the benefits of publishing this dataset should already provide enough advantages for the community to build upon in the future.
> > > >
> > > > Correspondingly, I have raised my score (from 6 to 7).
> > > >
> > > > As a remaining minor point that I would encourage the authors to consider for a final revised paper version: I would reconsider the placement of the related Hangul work paragraph. Whereas I appreciate that the authors have now included statements on related work in this direction, the present position of the respective paragraph in the introductory section feels somewhat misplaced/unintuitive. Specifically, it is placed after mention of some image datasets, but before the more general text on motivation, compositionality/disentanglement + the actual contribution statement for the Hangul dataset. My subjective preference would be to first have the motivation coherently in place (as has been done before) and state the contribution, before having a (potentially separate subsection) related work paragraph. Given that there is a lot of space left on the additional, last page, it certainly wouldn't hurt to further expand this text with a few sentences to further emphasise the differences to prior datasets (It's already there now, but it's very short and might be hard to follow for readers unfamiliar with these datasets).

---

### Official Review · Reviewer_uQCh · 2021-07-04
**Hangul fonts dataset is not general like ImageNet**

**Rating:** 6
**Confidence:** 1

**Strengths:**

1. the new dataset is helpful to analyze the hierarchy and compositionality in representation learning.

2. The experiments are comprehensive and the results are reasonable.

3. The dataset is synthetic, which means that it can be extended easily .

**Weaknesses:**

1. The part of sec 2 may be hard to read if someone is not familiar with Hangul.

2. The Hangul fonts dataset is not general or practical like ImageNet. There may be some limitations. For example, the conclusions with respect to hierarchy and compositionally based on this dataset may be not suitable for downstream works.

**Additional Feedback:**

-

**Clarity:**

The paper is well-organized, but may hard to read for those without some knowledge of Hangul fonts.

**Correctness:**

 The dataset is constructed in a sound way. The experiment is comprehensive and correct

**Documentation:**

The authors have provided the github repository for creating the dataset, which can extend the dataset easily if there are some special requirements.

**Ethics:**

There are no ethical concerns because the dataset is synthetic.

**Relation To Prior Work:**

Yes, the relation to prior works has been discussed clearly.

**Summary And Contributions:**

The authors proposed a new Hangul Fonts Dataset in order to address the problem of lacking hierarchy and compositionality in common datasets, such as ImageNet. With the dataset, research can investigate hierarchy and compositionality in representation learning methods. Meanwhile,  the authors provide several empirical investigations with respect to the hierarchical and compositional structure of the dataset, showing it is beneficial to the areas.

---

> ### Author Response · Authors · 2021-07-14
> **Detailed review response**
>
> We appreciate the constructive review and suggestions to improve clarity. We reply inline
>
> >Strengths:
>
> >1. the new dataset is helpful to analyze the hierarchy and compositionality in representation learning.
> >2. The experiments are comprehensive and the results are reasonable.
> >3. The dataset is synthetic, which means that it can be extended easily.
>
> We are grateful for the positive reviews of the dataset and example analyses.
>
> >Weaknesses:
>
> >1. The part of sec 2 may be hard to read if someone is not familiar with Hangul.
>
> We have revised Section 2 with a focus on clarity for readers unfamiliar with Hangul. We have moved the discussion of the linguistic meaning of blocks and glyphs to an appendix since it is not relevant for the annotations to streamline the description of the dataset. In addition, we have made changes to have more consistent and simplified terminology across the manuscript.
>
>
> >2. The Hangul fonts dataset is not general or practical like ImageNet. There may be some limitations. For example, the conclusions with respect to hierarchy and compositionally based on this dataset may be not suitable for downstream works.
>
> We appreciate the suggested clarifications. We have expanded on the section in the introduction where we compare the Hangul Fonts Dataset to other existing datasets and have clarified the types of structure each dataset may contain explicitly or implicitly.
> Additionally, we have expanded on the limitations of the dataset with respect to the expected generality of the results in the Discussion section:
> “The specific compositional and hierarchical structure in the HFD and the particular encodings and analysis used may not be applicable across all different types of compositionality or hierarchy, for instance some hierarchy may be fuzzy, rather than discrete and tree-like. However, we hope that the HFD inspires more research into tools for extracting these features from learned representations.”
>
> >Correctness:
>
> >The dataset is constructed in a sound way. The experiment is comprehensive and correct
>
> >Clarity:
>
> >The paper is well-organized, but may hard to read for those without some knowledge of Hangul fonts.
>
> We are glad to hear that you found the dataset and manuscript generally clear and correct. We have worked to increase the clarity of the manuscript for readers not familiar with Hangul, especially Section 2.

---

### Official Review · Reviewer_W4W1 · 2021-07-04
**Review on Submission 87**

**Rating:** 6
**Confidence:** 4

**Strengths:**

1. The explicit hierarchy and compositionality of the proposed dataset is beneficial for detailed investigations on representation learning.
2. The metric used for hierarchy and composition evaluation is quite interesting to me.
3. The experiments conducted are thorough and clear, supporting the conclusions well.
4. The dataset could be easily extended with different fonts. And the authors provide corresponding codes.

**Weaknesses:**

1. Section 2 is a little harsh to follow for those with little knowledge of Hangul.

2. In my opinion, it would be better if the authors could provide statistical analysis for the claims in L104-107. For example, the KMeans clustering accuracy with `Initial Label` as representation and `Medial Label` as the generative variable.

3. In Fig. 4C/D, there exists normalized accuracy lower than zero. To me it is unexpected since this implicates that the accuracy is lower than chance accuracy. Could the authors give some explanations on how did this happen?

4. I expect the KMeans clustering accuracy between different generative variables. I think the result could demonstrate the upperbound of the hierarchical structure information that could be learned by supervised methods, which could justify why the hierarchical structure information should be acquired by the representation learned by supervised methods.

5. It seems except the analysis involving 'all geometry', few analysis on representation and generative variables from different levels is conducted. I would love to see the Bag-of-Atoms features to be contained in the analysis in Fig. 4.

**Additional Feedback:**

1. In Line 95, it seems that a period is missed before ``` 'There'.

**Clarity:**

Though Section 2 is harsh to follow for those with little knowledge of Hangul, the other parts are generally clear.

**Correctness:**

I believe the claims are correct. The dataset is constructed soundly. The experiment is well-designed and appropriate.

**Documentation:**

The data collection and organization, availability and maintenance, and ethical and responsible use are sufficiently provided.

**Ethics:**

There are no ethical concerns that warrant further discussion or review.

**Relation To Prior Work:**

The relation to prior works has been clearly discussed.

**Summary And Contributions:**

To advance the investigation on hierarchy and compositionality in representation learning methods, the authors proposed the Hangul Fonts Dataset, which is of known hierarchy, compositional structure and naturalistic variability. Several typical representation learning methods are evaluated on the dataset. Analysis on both unsupervised and supervised methods are given.

---

**Updated after the authors‘ response:**

The authors addressed my main concerns in the rebuttal thoroughly. Though there are still open issues, I believe publishing this work would encourage the community to work on these. Therefore, I raise my rating from 5 to 6.

---

> ### Author Response · Authors · 2021-07-14
> **Detailed review response**
>
> We thank the reviewer for the constructive review. We respond to the comments inline
>
> >Strengths:
>
> >1. The explicit hierarchy and compositionality of the proposed dataset is beneficial for detailed investigations on representation learning.
> >2. The metric used for hierarchy and composition evaluation is quite interesting to me.
> >3. The experiments conducted are thorough and clear, supporting the conclusions well.
> >4. The dataset could be easily extended with different fonts. And the authors provide corresponding codes.
>
> We are glad that the reviewer found the dataset and analysis interesting and the manuscript generally clear.
>
> continued...

---

> > ### Author Response · Authors · 2021-07-14
> > **Details review response continued**
> >
> > >Weaknesses:
> > >1. Section 2 is a little harsh to follow for those with little knowledge of Hangul.
> >
> > We have revised Section 2 with a focus on clarity for readers unfamiliar with Hangul. We have moved the discussion of the linguistic meaning of blocks and glyphs to an appendix since it is not relevant for the annotations to streamline the description of the dataset. In addition, we have made changes to have more consistent and simplified terminology across the manuscript.
> >
> >
> > >2. In my opinion, it would be better if the authors could provide statistical analysis for the claims in L104-107. For example, the KMeans clustering accuracy with Initial Label as representation and Medial Label as the generative variable.
> >
> > We appreciate this suggestion. We have added the following quantification of the claims about deep network recovery of the contextual dependence of the initial glyph on the medial geometry to the third paragraph of Section 3.
> > “The medial\_geometry variable can be decoded with accuracy significantly above chance across all layers (p $<.01$, 1-sample t-test). However, the normalized accuracy drops from about 0.22 in the first layer to less than .01 by the last layer. This indicates that although the network may be using the medial geometry context in the early layers, it is compressed out of the representation by the final layers.”
> >
> > >3. In Fig. 4C/D, there exists normalized accuracy lower than zero. To me it is unexpected since this implicates that the accuracy is lower than chance accuracy. Could the authors give some explanations on how did this happen?
> >
> > It is true that several of the mean normalized accuracies in Fig 4 are lower than chance. Although it is potentially interesting that they all happen for Initial Geometry, the negative values are not all significantly different from 0 and even the ones that are significant have pvalues > .01 (1 sample t-test). In addition, the differences from 0 are relatively small. Furthermore, inspecting the per-fold accuracies shows that it was just one or two of the 7 folds that had a larger below chance accuracy. Given this, we would attribute this to statistical fluctuations rather than a meaningful signal.
> > This explanation has been added to the corresponding paragraph in the text:
> > “There are several mean normalized accuracies that are less than zero. Although it is potentially interesting that it only occurs for Initial Geometry, the negative values all have pvalues $> .01$ (1 sample t-test) and some are not significantly different from 0. In addition, the differences from 0 are relatively small. Furthermore, inspecting the per-fold accuracies shows that it was just one or two of the 7 folds that had a larger below chance accuracy.”
> >
> >
> > continued...

---

> > > ### Author Response · Authors · 2021-07-14
> > > **Detailed reivew response continued**
> > >
> > > >4. I expect the KMeans clustering accuracy between different generative variables. [...]
> > >
> > > We agree that the relationship between different sets of generative variables can potentially influence the accuracy of recovery between variables. Indeed, understanding whether representation learning methods organize their representations around generative variables they are not directly trained on is a main motivation for the dataset and manuscript. To your particular point, an example would be that training on the “Initial” class label may automatically structure the representations around the “Initial Geometry” labels since they are partially correlated. It is not clear to us that it necessarily provides an upper (or lower) bound. For example, if a network is trained on the “Initial” classes and learns a highly clustered representation for each class, it is not guaranteed the network will always put classes that share “Initial Geometry” hierarchy close to each other in the space (this is a type of hypothesis we are hoping to test with the dataset). This could result in cluster accuracies lower than what was expected based on the label correlations. Similarly, the network could perfectly group “Initial” class representations around their “Initial Geometry” labels and the clustering accuracy would be limited by the “Initial” accuracy with some conversion to account for different numbers of classes.
> > > Given our current limited understanding of how to account for this, we have not added any quantitative analysis around these questions. We have expanded on the relationships between different sets of generative variables in Section 2 which reads:
> > > “These three sets of variables---IMF class labels, hierarchy class labels, and bag-of-atoms binary features---are not independent of each other. For example, training on the Initial class label may automatically structure the learned representations around the Initial Geometry labels since they are partially correlated. However, it is not clear whether this provides an upper (or lower) bound for the expected structure of related variables in the representation. For example, if a network is trained on the Initial classes and learns a highly clustered representation for each class, it is not guaranteed the network will always put classes that share Initial Geometry hierarchy close to each other in the learned representations. Indeed, this is a hypothesis we are hoping to test with this dataset across representation learning methods. This could result in clustering accuracies lower than what was expected based on the label correlations. Similarly, the network could perfectly group Initial class representations around their Initial Geometry labels and the clustering accuracy would be set by the Initial accuracy with some conversion to account for different numbers of classes.”
> > >
> > >
> > > >5. It seems except the analysis involving 'all geometry', few analysis on representation and generative variables from different levels is conducted. I would love to see the Bag-of-Atoms features to be contained in the analysis in Fig. 4.
> > >
> > > We agree that the analysis of hierarchy is limited due to the shallow hierarchy present. The reason the Bag-of-Atoms features were not included in the KMeans cluster analysis in Figure 4 is that we could not think of a way to compare the KMeans clustering label, which is a label/one-hot vector, with the binary feature vectors from the Bag-of-Atoms representation. Each sample can only be identified with one cluster centroid, but it may have a variable number of features present. For the hierarchy variables, there is a clear way to associate them with a KMeans clustering since they are also labels/one-hot vectors.
> > > We have added the following text in the Discussion to address this limitation:
> > > “The specific compositional and hierarchical structure in the HFD and the particular encodings used may not be applicable across all different types of compositionality or hierarchy, for instance some hierarchy may be fuzzy, rather than discrete and tree-like. Similarly, the analysis presented here is tailored to the particular structures present in the data. For example, the KMeans clustering analysis was applied to all variables with mutually-exclusive class structure (i.e., one cluster centroid), but could not be applied to the bag-of-atoms feature vectors since a sample may have a variable number of features present. However, we hope that the HFD inspires more research into tools for extracting these features from learned representations.”
> > >
> > > >Clarity:
> > >
> > > >Though Section 2 is harsh to follow for those with little knowledge of Hangul, the other parts are generally clear.
> > >
> > >  We are happy to hear that the remainder of the manuscript is clear. We have made changes to Section 2.
> > >
> > > >Additional Feedback:
> > >
> > > >In Line 95, it seems that a period is missed before ``` 'There'.
> > >
> > > We have fixed this typo.

---

### Author Response · Authors · 2021-07-14
**General response to all reviewers**

We are grateful for the constructive reviews from the 3 reviewers. We believe that we have addressed concerns raised which has strengthened the manuscript. The main concerns we addressed are
1. A lack of clarity in Section 2 which describes the structure of the Hangul Fonts Dataset
2. Missing code for the example analysis section (Section 3) of the manuscript
3. Limitations which were not addressed in the manuscript

For 1., we have revised Section 2 with a focus on clarity for readers who may not be familiar with the Hangul writing system. We have moved the discussion of linguistic meaning of blocks and glyphs to the Appendix and have been careful to use consistent terminology. For 2., we have included the license and analysis code in the Hangul Fonts Dataset repository. We have also created a Zenodo repository that contains the version of the code used in the manuscript. We have also better addressed several limitations the reviewers commented on in the Discussion section.

We will respond to the detailed review comments in the respective threads.

---

### Decision · Program_Chairs · 2021-07-27

**Decision:**

Reject

**Comment:**

This paper received borderline to positive reviews. However, after considering the points raised by the reviewers and reading the paper, the AC does not think the paper is above the bar of NeurIPS. The dataset itself is artificial and not well motivated, as it does not directly serve a practical application. Given this, it is not clear why Hangul fonts are special, and why they are preferred over composing some arbitrary shapes.

In addition, the utility of a dataset (benchmark) is partially dependent on a standardized evaluation protocol. This paper is lacking in this aspect. Although the paper has performed analysis, it is not clear what is a well-justified protocol and metric in terms of evaluation disparate methods in terms of hierarchy and composition. In general, the protocol and metric should be simple and interpretable, while kmeans on the learn representations may shed some light, it is not clear how rigorous the conclusions are.